# Wealth-related inequality in vitamin A rich food consumption among children of age 6–23 months in Ethiopia; Wagstaff decomposition of the 2019 mini-DHS data

**Mehari Woldemariam Merid**[ORCID][1]*, **Fantu Mamo Aragaw**[1], **Tilahun Nega Godana**[2], **Anteneh Ayelign Kibret**[3], **Adugnaw Zeleke Alem**[1], **Melaku Hunie Asratie**[4], **Dagmawi Chilot**[5], **Daniel Gashaneh Belay**[3]

1 Department of Epidemiology and Biostatistics, Institute of Public Health, College of Medicine and Health Sciences, University of Gondar, Gondar, Ethiopia, 2 Department of Internal Medicine, University of Gondar College of Medicine and Health Science, Comprehensive Specialized Hospital, Gondar, Ethiopia, 3 Department of Human Anatomy, College of Medicine and Health Sciences, University of Gondar, Gondar, Ethiopia, 4 Department of Women's and Family Health, School of Midwifery, College of Medicine and Health Sciences, University of Gondar, Gondar, Ethiopia, 5 Department of Human Physiology, College of Medicine and Health Sciences, University of Gondar, Gondar, Ethiopia

* mehariho19@gmail.com

## Abstract

### Introduction

Vitamin A (VA) cannot be made in the human body and thus foods rich in VA are the only sources of vitamin A for the body. However, ensuring availability in adequate amount of foods rich in VA remains a challenge, mainly in low-income counties including Ethiopia. In addition, children from the poorest and less educated families of same country have disproportionately limited consumptions of foods rich in VA. Therefore, the present study aimed assessing the wealth related inequality in vitamin A consumption (VAC) and decompose it to the various contributing factors.

### Methods

This study was conducted using the 2019 Ethiopian demographic and health survey data on a weighted sample of 1,497 children of age 6–23 months in Ethiopia. The wealth related inequality in VAC was quantified using concentration index and plotted using concentration curve. The Wagstaff decomposition analysis was performed to assess the relative contributions of each explanatory variable to the inequalities in the overall concentration index of VAC.

### Result

The overall Wagstaff normalized concentration index (C) analyses of the wealth-related inequality in consumption of foods rich in VA among children aged 6–23 months was [C = 0.25; 95% C: 0.15, 0.35]. Further decomposition of the C by the explanatory variables

**Data Availability Statement:** Third party data was obtained for this study from The DHS Program (https://dhsprogram.com/). Data may be requested

from The DHS Program after creating an account and submitting a concept note. More access information can be found on The DHS Program website (https://dhsprogram.com/data/Access-Instructions.cfm).The data set is openly available upon permission from the MEASURE DHS website (https://www.dhsprogram.com/data/available-datasets.cfm). The authors confirm that interested researchers would be able to access these data in the same manner as the authors. The authors also confirm that they had no special access privileges that others would not have.

**Funding:** The author(s) received no specific funding for this work.

**Competing interests:** The authors have declared that no competing interests exist.

**Abbreviations:** ANC, antenatal care; BF, breast feeding; C, concentration index; DHS, Demographic and Health Surveys; EAs, Enumeration Areas; EMDHS, Ethiopian mini Demographic and Health Survey; PNC, postnatal care; VAC, vitamin A consumption; VA, Vitamin A; WHO, World Health Organization.

reported the following contributions; primary level of women's education (7.2%), secondary and above (17.8%), having ANC visit during pregnancy (62.1%), delivery at a health institution (26.53%), living in the metropolis (13.7%), central region (34.2%), child age 18–23 months (4.7%) contributed to the observed wealth related inequality in the consumption of foods rich in vitamin A in Ethiopia.

## Conclusion

We found pro-rich wealth-related inequality in VAC among children of age 6–23 months in Ethiopia. Additionally, maternal education, region, ANC visit, and place of delivery were the significant contributors of wealth-related inequality of VAC. Nutritional related interventions should prioritise children from poorer households and less educated mothers. Moreover, enhancing access to ANC and health facilities delivery services through education, advocacy, and campaign programs is highly recommended in the study setting.

## Introduction

Inadequate micro-nutrients feeding practices to infants and young children, particularly in the first 2 years of life, are the key risk factors for under-nutrition and delayed cognitive development and growth mainly in developing countries [1]. Good VA consumption refers to the intake of foods rich in VA at least one food item among the main type of food items [2].

The human body cannot produce VA and thus foods rich in it are the only sources for the human body demand of vitamin A [2]. However, ensuring that VA is available in adequate amount remains a challenge, mainly in resource-limited countries including Ethiopia. Globally, it is estimated that more than 124 million children require access to VA rich foods or supplementations [3]. In most African countries, the dietary intake of Vitamin A-rich foods and other micronutrients is below the daily-recommended amount [4–6]. In Ethiopia, about 62% of children aged 6–23 months had poor vitamin A food intake [7]. In most low and middle-income countries including Ethiopia, there has been disproportionately high burden of vitamin A deficiency (VAD) and limited consumption of foods rich in VA among children in the poorest segment of the population [7, 8].

VAD, a common form of micronutrient deficiency [9], has been magnified by poverty status and higher prevalence of infectious diseases [10], and it is the underlying cause of morbidity and mortality due to measles, diarrhea, and malaria among childeren of age 6–59 months in African countries, including Ethiopia [3, 4, 11, 12]. Furthermore, VAD causes night blindness problem, Bitot's spots and other morbid conditions [13]. The high rate of VAD is partly due to poor consumption of foods rich in VA [14].

Some literatures noted the existence of socio-economic related inequalities in access to micronutrients and the contributing factors to the inequality among children in low resource countries [15–18]. The concentration index and the Wagstaff decomposition analyses are the appropriate statistical approaches to estimate the extent of inequality of a particular health outcome and identify the possible contributing factors for the observed inequality of that outcome [19]. Accordingly, women's level of education, income, age, marital status, occupation, place of residence, use of prenatal and antenatal care, child age, sex, current breast-feeding status, birth order, and number of under five children in the family are some of the key contributors to the socio-economic related inequalities to access to VAC in particular and micro-nutrients

in general [8, 17, 18, 20, 21]. Hence, identifying and reducing avoidable contributors of socio-economic inequalities to VA rich foods is a critical step towards improving children's overall health and well-being [16].

Micronutrient deficiencies including the VAD can be effectively prevented through alternative approaches including supplementation programs, and dietary adjustments, and other local contexts [11, 22, 23]. Notably, Vitamin A Supplementation is an effective strategy that shown remarkable reduction in the morbidity and mortality related to vitamin A deficiency disorders in different countries [24, 25]. However, it does not sustainably improve the vitamin A status of a population for it helps only to attain minimally adequate vitamin A stores during the first 2 years of life [23].Thus, it is imperative to consider dietary modifications that leads to the consumption of vitamin A-rich foods as the most appropriate measure to control the burden of VAD [26].

Although literatures are available regarding the determinants of consumptions of foods rich in vitamin A [14, 27–30], we couldn't locate, to the best of our search, a study that assessed wealth related inequality and contributors of vitamin A consumption among children in Ethiopia. The findings of this study may inform the formulation of policies and actions and would help to prioritise targeted interventions to reduce the existing wealth related inequalities of VAC in Ethiopia.

Therefore, the present study assessed the wealth related inequality in VAC and decompose it to the various socio-demographic and economic, maternal health care utilization, and child characteristics that may contribute to the overall inequality in VAC in Ethiopia.

## Method

### Data sources and populations

The present study was conducted using the 2019 Ethiopian Mini Demographic and Health Survey (EMDHS) dataset. It is the fifth DHS implemented in Ethiopia from March 21, 2019 to June 28, 2019. The 2019 EMDHS was conducted by the Central Statistical Agency in partnership with the Federal Ministry of Health and the Ethiopian Public Health Institute.

The sample used for the survey was stratified and selected using two stages. Firstly, a total of 305 EAs (93 in urban, 212 in rural) were chosen independently with a probability proportional to each EAs. Second, from the newly formed household listing, a fixed number of 30 households/clusters were selected with an equal probability of systematic selection. The detailed sampling procedures are available on the measure DHS website in the 2019 EMDHS report accessible on (https://www.dhsprogram.com). Third party data was obtained for this study from The DHS Program (https://dhsprogram.com/). Data may be requested from The DHS Program after creating an account and submitting a concept note. More access information can be found on The DHS Program website (https://dhsprogram.com/data/Access-Instructions.cfm).The data set is openly available upon permission from the MEASURE DHS website (https://www.dhsprogram.com/data/available-datasets.cfm). The authors confirm that interested researchers would be able to access these data in the same manner as the authors. The authors also confirm that they had no special access privileges that others would not have The kids record (KR) data set was used to include a total weighted sample of 1,497 infants and young children of age 6–23 months in the current study.

### Variables of the study

**Outcome variable.**   In this study the outcome variable was wealth-related inequality in consumption of foods rich in vitamin A among infants and young children aged 6–23 months living with their mothers/caregivers who have taken at least one food item among the seven

food items; 1) Have the child taken eggs? 2) Have the child taken meat (beef, pork, lamb, chicken, etc.)? 3) Have the child taken a pumpkin, carrots, and squash (yellow or orange inside)? 4) Have the child taken any dark green leafy vegetables? 5) Have the child taken mangoes, papayas, and another vitamin A fruit? 6) Have the child was taken liver, heart, and other organs? 7) Have the child taken fish or shellfish?) at any time in the last 24 hours preceding the interview was considered good consumption of foods rich in vitamin A coded as "1", otherwise, no consumption coded as "0" [31]. These questions were asked to the mothers' or the caregivers' of the infants any time in the last 24 hours preceding the survey.

## Contributing variables to the inequality in vitamin A consumption

The contributing factors of VAC include women's level of education, age, marital status, occupation, place of residence, region, use of prenatal and antenatal care, child age, sex, current breast feeding status, birth order, and number of under five children in the family, family size, and household wealth index, were some of the key contributors to the socio-economicrelated inequalities to consumption of foods rich in vitamin A (Table 1).

**Operational definations.** The following varibles used in the present study were defined and categorized operationally based on the previous similar evidences available [32–34] and the DHS guideline [35].

**Distance to health facility.** Recorded as a big problem and not a big problem in the dataset was taken without change, which is respondents' perception during the survey whether they perceived the distance from their home to the nearest health facility to get self-medical help as a big problem or not [36].

**Wealth index.** In the DHS dataset, wealth index was created using principal components analysis coded as "poorest", "poorer", "Middle", "Richer", and "Richest and taken as it is.

**ANC visit.** Was measured in number and mothers who had at least one ANC visit was considered as having ANC visit [37].

**Region.** Was coded into nine regions and two city administration: Tigray, Afar, Amhara, Oromiya, Somali, Benishangul-Gumuz, South Nation and Nationalities People (SNNP), Gambela, Harari, Addis Ababa, and Dire Dawa. We further categorized the region in three categories as; central (Tigray, Amhara, Oromia, and SNNP), metropolis (Addis Ababa, Dire Dawa, and Harari), and periphery (Afar, Somali, and Gambella).

**Occupation.** Occupational status of mother was categorized as currently working (includes all types of work) and currently not working based on the respondents response at the time of the survey.

**Breast feeding status.** This has been assessed by asking the mothers and recoreded as "yes" if they were breast fed their infant/child during the survey period and "no" otherwise [36].

## Data management and analysis

Data extraction, editing, coding, and cleaning were performed using STATA 14 software. The data was weighted using sampling weight (v005) to restore the survey's representativeness and obtain valid statistical estimates. The wealth related inequalities in VAC were quantified using concentration index (C) and depicted using concentration curves (CC). The concentration curve is obtained by plotting the cumulative proportion of the outcome variable (VAC) on y-axis against the increasing percentage of the population ranked by the household wealth index (poorest to richest) on x-axis. If the curve is above the line of equality (45 degree line) that means the index value is negative hence it shows that the outcome variable is disproportionally concentrated among the poor and vice-versa.

**Table 1. Socio-demographic and economic, maternal and childhood characteristics by consumption of foods rich in vitamin A in Ethiopia; mini EDHS 2019 [N = 1497].**

| Variables | Categories | Vitamin A food consumption | |
|---|---|---|---|
| | | Good (%) | No (%) |
| Women's age | 15–24 | 184 (38.75) | 290 (61.25) |
| | 24–34 | 315 (41.85) | 437 (58.15) |
| | 35–49 | 87 (32.15) | 184 (67.85) |
| Marital status | Single | 37 (44.21) | 47 (55.79) |
| | Married | 548 (38.81) | 865 (61.19) |
| Household wealth index | Poorest | 58 (19.51) | 240 (80.49) |
| | Poorer | 121 (38.54) | 194 (61.46) |
| | Middle | 106 (37.57) | 176 (62.43) |
| | Richer | 121 (45.89) | 143 (54.11) |
| | Richest | 179 (52.93) | 159 (47.07) |
| U-5 children | 1 | 256 (40.33) | 378 (59.67) |
| | 2 | 298 (42.23) | 407 (57.77) |
| | > = 3 | 32 (20.38) | 126 (79.62) |
| Mothers educational statusMothers educational status | No education | 194 (29.17) | 472 (78.83) |
| | Primary | 280 (44.95) | 343 (55.05) |
| | Secondary & above | 111 (53.43) | 97 (46.57) |
| Residence | Urban | 204 (47.75) | 224(52.25) |
| | Rural | 381 (35.65) | 688 (64.35) |
| Region | Metropolitans | 34 (54.81) | 28 (45.19) |
| | Central | 535 (41.20) | 763 (58.80) |
| | Periphery | 17 (12.34) | 120 (87.76) |
| Had ANC visit | Yes | 483 (43.47) | 628 (56.53) |
| | No | 102 (26.60) | 284 (73.48) |
| Place of delivery | Home | 230 (35.35) | 420 (64.65) |
| | Health facility | 356 (42.00) | 491 (58.00) |
| Had PNC | Yes | 88 (44.64) | 110 (55.36) |
| | No | 497 (38.27) | 802 (61.73) |
| Age in month | 6–11 | 148 (30.73) | 333 (69.27) |
| | 12–17 | 205 (36.84) | 352 (63.16) |
| | 18–23 | 232 (50.64) | 227 (49.36) |
| Sex | Male | 295 (38.18) | 478(61.82) |
| | Female | 290 (40.10) | 434 (59.90) |
| Birth order | 1st | 156 (42.83) | 208 (57.17) |
| | 2nd to 4th | 305 (43.16) | 401 (56.84) |
| | > = 5th | 125 (29.24) | 302 (70.76) |
| Currently BF | Yes | 501 (39.36) | 771 (60.64) |
| | No | 85 (37.70) | 140 (62.30) |

**ANC**-antenatal care, **BF**-breast feeding, **C**–concentration index, **PNC**-postnatal care

The C is a measure of inequality in the distribution of health outcome across the wealth distributions, which reflects the experience of the population as the whole and is sensitive to the change in the distribution of the population across socio-economic groups and can be decomposed. Its value ranges from − 1 and + 1, where C = 0 shows perfect equality while C < 0

indicates VAC is disproportionately concentrated among the poor (pro-poor inequality), and C>0 means VAC is disproportionately concentrated among the rich (pro-rich inequality) [19].

### Decomposition of the concentration index (C)

To assess the relative contributions of each factor to inequalities in VAC, we decomposed the concentration index of VAC into its contributory factors using the Wagstaff approach [38].

Using this method of decomposition, we first calculated the concentration index of VAC and then each of the contributors to the concentration index. Second, we calculated the absolute contribution of each factor (*x*) to the concentration index of VAC (*y*). The concentration index could be decomposed into the contributions of each factor to the wealth-related inequalities [39]. For any linear regression model on a health outcome (y), such as;

$$y = \propto + \sum\nolimits_k \beta_k x_k + \varepsilon \qquad (1)$$

The concentration index for y is then written as;

$$C = \sum\nolimits_k (\beta_k X_k / \mu) C_k + G C_\varepsilon / \mu) \qquad (2)$$

Where $\mu$ is the mean of $y$, $X_k$ is the mean of $x_k$, $C_k$ is the concentration index for $x_k$, and $GC_\varepsilon$ is the generalized concentration index for the error term $\varepsilon$. Eq (2) shows that $C$ is equal to a weighted sum of the concentration indices of the $k$ regressor, where the weight for $x_k$ is the elasticity of $y$ with respect to $x_k$. The residual component captured by the error term reflects the wealth-related inequality in health that is not explained by systematic variation in the regressor by wealth, which should approach zero for a well-specified model. Each contribution is the product of elasticity with the degree of economic inequality. Moreover, the percentage contribution is obtained by dividing each absolute contribution by total absolute contribution multiplied by 100 to obtain the estimates [40].

The decomposition analyses of the concentration index showed the relative contribution of each variable to the inequalities in VAC, as shown in Table 2. The table presents the concentration index for each of the variables, the elasticity, the absolute contribution as well as the percentage contribution of each variable to the inequalities in VAC. The negative sign in the C indicates the more concentration of VAC among the poor, where a positive value indicates concentration among the rich. The elasticity indicates the sensitivity of VAC for each of the contributing factors. Absolute contribution is calculated by multiplying elasticity with the C of each factor and indicates the extent of inequality contributed by the explanatory variables. Whereas, percent contribution refers to the contribution of each variable to the overall concentration index.

### Ethical consideration

All methods were carried out following relevant guidelines of the Demographic and Health Surveys (DHS) program. Informed consent was waived from the International Review Board of Demographic and Health Surveys (DHS) program data archivists. After the consent paper was submitted to DHS Program, a letter of permission to download the dataset was obtained for this study. The dataset was not shared or passed on to other bodies and was anonymized to maintain its confidentiality.

**Table 2. Decomposition of the concentration index of wealth-related inequalities to VAC attributable to maternal and child factors among children aged 6–23 months in Ethiopia; mini EDHS 2019.**

| Variables | Categories | Coefficient | [95% C] | Elasticity | C | Cont.a | %Cont.b |
|---|---|---|---|---|---|---|---|
| **Socio-demographic and economic factors (sub-total)** | | | | | | | **28.884** |
| Age in years | 15–24 (**reff**) | | | | | | |
| | 24–34 | 0.0144 | -0.0549, 0.0836 | 0.1248 | 0.040 | 0.005 | 1.983 |
| | 35–49 | -0.0137 | -0.1157, 0.0882 | 0.0193 | -0.056 | -0.0011 | -0.441 |
| Marital status | Single (**reff**) | | | | | | |
| | Married | -0.0124 | -0.1171, 0.0923 | -0.123 | 0.009 | -0.001 | -0.435 |
| U-5 children | 1 | 0.0929 | -0.0023, 0.1883 | 0.1581 | 0.200 | 0.032 | 12.495 |
| | 2 | 0.0997 | **0.0153, 0.1841** | 0.254 | -0.096 | -0.024 | **-9.672** |
| | > = 3 (**reff**) | | | | | | |
| Mothers educational status | No education (**reff**) | | | | | | |
| | Primary | 0.1306 | **0.0671, 0.1942** | 0.1603 | 0.113 | 0.018 | **7.160** |
| | Secondary & above | 0.2534 | **0.1675, 0.3394** | 0.0698 | 0.645 | 0.045 | **17.794** |
| **Geographic factors (sub-total)** | | | | | | | **68.533** |
| Residence | Urban | 0.0457 | -0.0237, 0.1151 | 0.076 | 0.685 | 0.052 | 20.706 |
| | Rural (**reff**) | | | | | | |
| Region | Metropolitans | 0.0931 | **0.0141, 0.1721** | 0.049 | 0.712 | 0.034 | **13.679** |
| | Central | 0.1048 | **0.0443, 0.1655** | 0.934 | 0.093 | 0.086 | **34.151** |
| | Periphery (**reff**) | | | | | | |
| **Maternal health care utilization (sub-total)** | | | | | | | **88.603** |
| Had ANC visit | Yes | 0.0704 | **0.0039, 0.1369** | 0.357 | 0.440 | 0.157 | **62.138** |
| | No (**reff**) | | | | | | |
| Place of delivery | Home (**reff**) | | | | | | |
| | Health facility | 0.0954 | **0.0343, 0.1565** | 0.129 | 0.521 | 0.067 | **26.527** |
| Had PNC | Yes | 0.0083 | -0.0679, 0.0845 | 0.0068 | 0.108 | 0.0007 | 0.292 |
| | No (**reff**) | | | | | | |
| **Child characteristics (sub-total)** | | | | | | | **14.225** |
| Age in month | 6-11(**reff**) | | | | | | |
| | 12–17 | 0.0742 | **0.0116, 0.1367** | 0.083 | -0.029 | -0.002 | **-0.949** |
| | 18–23 | 0.1827 | **0.1147, 0.2507** | 0.219 | 0.054 | 0.012 | **4.659** |
| Sex | Male | 0.0004 | -0.0503, 0.0511 | -0.013 | -0.006 | 0.0001 | 0.034 |
| | Female (**reff**) | | | | | | |
| Birth order | 1st | -.0376 | -0.1444, 0.0693 | 0.087 | 0.244 | 0.021 | 8.3851 |
| | 2nd to 4th | -.0391 | -0.1103, 0.0321 | 0.147 | 0.026 | 0.004 | 1.527 |
| | > = 5th (**reff**) | | | | | | |
| Currently BF | Yes | 0.0345 | -0.0342 0.1032 | 0.058 | 0.025 | 0.001 | 0.569 |
| | No (**reff**) | | | | | | |

**ANC**-antenatal care, **BF**-breast feeding, **C**–concentration index, **PNC**-postnatal care, **reff**-reference category,

Cont.**a** = Contribution to concentration index = C *Elasticity

%Cont.**b** = Percentage contribution to concentration index = (Cont.C/ Overall conc. index)*100

## Results

### Description of vitamin A consumption by the maternal and child characteristics

In this study, a total weighted sample of 1,497 children of age 6–23 was included. Nearly equal proportions of males (38.18%) and females (40.10%) consumed foods rich in vitamin A (VA).

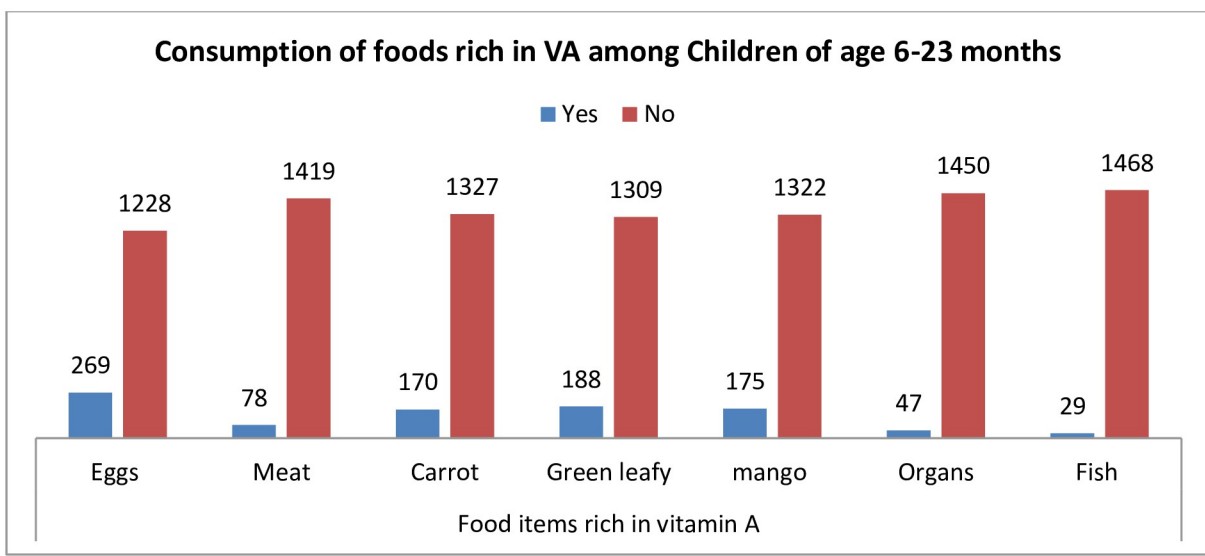

**Fig 1. Proportions of the consumption of food items rich in VA by children of age 6–23 months in Ethiopia.**

Just half (50.64%) of the children aged 18–23 months had good consumptions of foods rich in VA. With regard to educational status, only (29.17%) children born from mothers with no education were able to consumed foods rich in VA. About one-fifth (19.51%) of the children born from women lived in the poorest wealth quintile household had good consumption of foods rich in VA. Alternatively, above half (52.93%) of the children lived with women from the household labelled as the richest wealth quintile had good consumption of VA (Table 1).

## Consumptions of foods rich in vitamin A

Among the seven vitamin A rich food items, the most commonly consumed item was egg, 269 (17.97%), and the least item consumed by the children was fish and shellfish foods, 29 (1.94%) [Fig 1].

## Wealth related inequality of VAC

The value of the C is ranged from -1 to 1. A negative sign indicates the more concentration of consumption of foods rich in vitamin A among the poor, whereas a positive value indicates concentration among the rich. In our study, the overall Wagstaff normalized concentration index analyses of the wealth-related inequality of VAC was [C = 0.25; 95% C: 0.15, 0.35]. The result showed that VAC among children aged 6–23 months was disproportionately concentrated on the richer groups which implied a pro-rich distribution of VAC. The concentration curve lies below the 45-degree diagonal line (line of equality) that indicates a higher level of consumption of VA rich foods concentrated among children from the rich socio-economicstatus (Fig 2).

## Wealth related inequality of VAC plotted by key variables

The concentration curve has further been used to show the inequality in VAC among children between or among categories of key variables in our study. As shown in Fig 3, the concentration index among women with no education was (C = -0.42; 95% C: -0.50, -0.35), (C = 0.11; 95% C: 0.03, 0.20) for primary education, and (C = 0.65; 95% C: 0.50, 0.80) for seccondary and

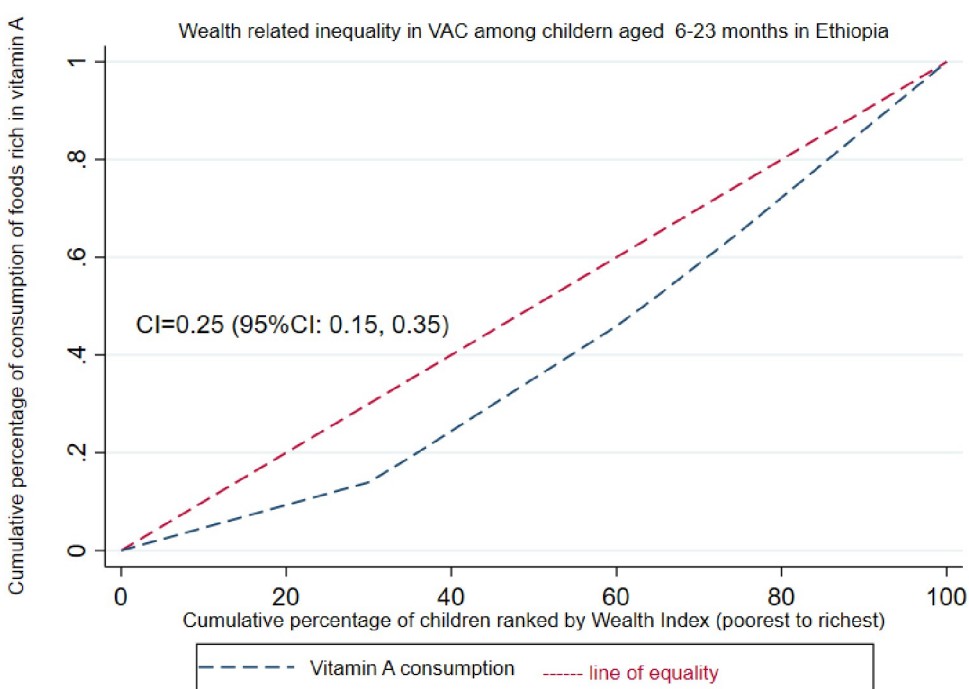

**Fig 2. Concentration curve showing the wealth related inequality in consumption of foods rich in vitamin A among children aged 6–23 months in Ethiopia.**

above level of education. Thus the tendency to consum foods rich in vitamin A among children lived with women who attained higher level of education was discrimatetely distributed towards the rich households (pro-rich distribution). Similarly the concentration curve was ploted for ANC visit and place of delivery as indicated in Fig 4.

## Decomposition of wealth-related inequality in VAC

We decomposed the VAC concentration index against the socio-demographic, socio-economic, and maternal and child related variables to show the relative contribution of each variable to inequalities in VAC.

Accordingly, the overall concentration index [0.25 (95% C: 0.15, 0.35; p = 0.01)] of VAC was further decomposed and explained by each explanatory variables as follows; more than one-fourth (28.9%) of the observed wealth related inequality in the consumption of vitamin A rich foods among children age 6–23 months was explained by the socio-demographic factors. The women's educational status; primary education (7.2%), secondary and above (17.8%), and the number of under five children in the household; one child (12.5%), two and above (-9.7%) were the socio-demographic significant contributors to the wealth related inequality to VAC. Additionally, having ANC visit during pregnancy and delivery at a health institution contributed about 62.1%, and 26.53% to the inequality in VAC, respectively. By geography, while living in the metropolis explained 13.7% to the inequality, central region contributed about 34.2% to the same. Lastly, age of the children (18–23 months) explained 4.7% to the observed wealth related inequality in the consumption of foods rich in vitamin A in Ethiopia (Table 2).

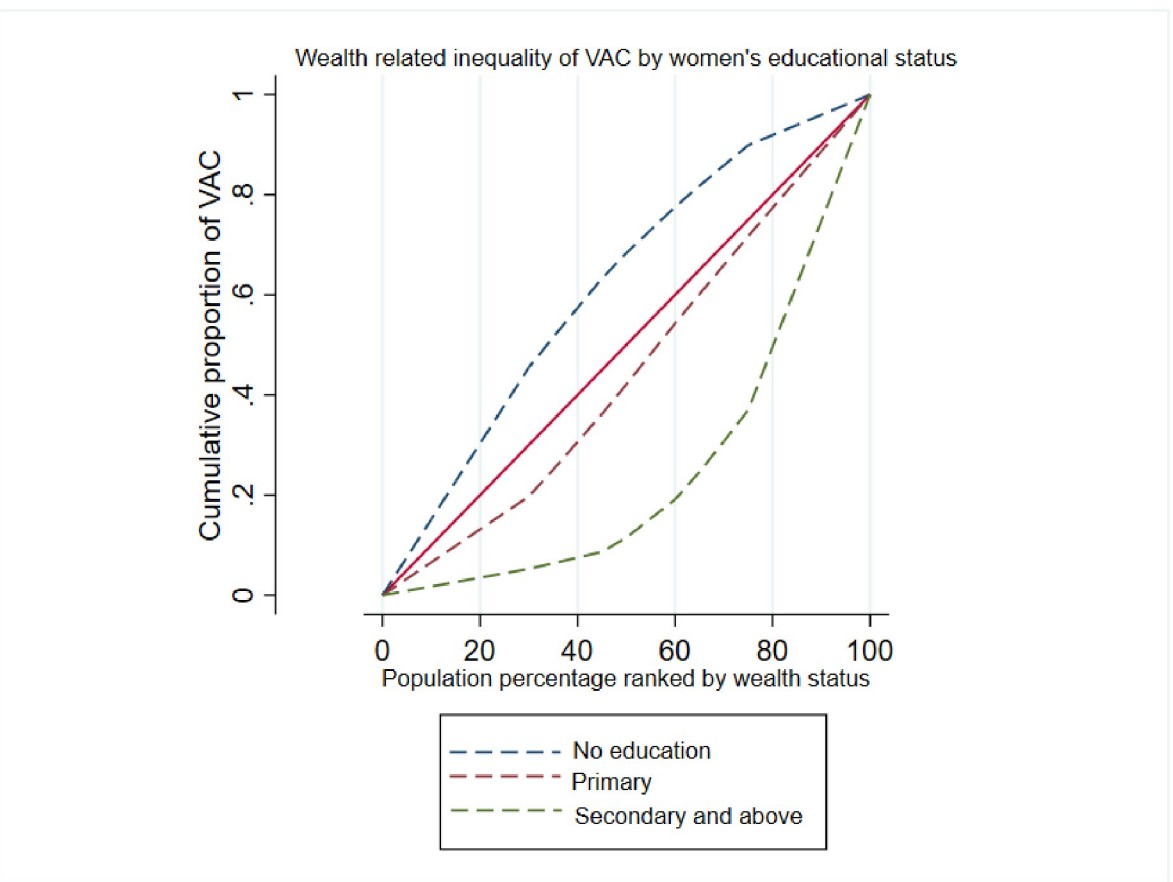

**Fig 3. Wealth related inequality to VAC among children attributable to women's educational status.**

## Discussion

In our study, the overall Wagstaff normalized concentration index analyses of the wealth-related inequality in consumption of foods rich in vitamin A (VAC, hereafter) was [C = 0.25; 95% C: 0.15, 0.35]. The result showed that VAC among children aged 6–23 months was disproportionately concentrated on the richer groups (pro-rich distribution), disfavouring the poor. Although directly not related, there have been some studies conducted on the socio-economic related inequality on minimum acceptable diet and recommended micronutrient intake among children of age 6–59 months in general [17, 18, 41, 42]. The findings of the aforementioned studies supported our study in that access to and consumption of foods rich in the micro-nutrients was discriminately concentrated in the rich households.

The above wealth-health association could be explained in that women from higher-income households could have accessed and were more likely to afford to have diversified foods including fruits and vegetables rich in vitamin A [43, 44]. As a result, children from such families might have good consumption of vitamin A rich foods compared to those children from a low household income [45].

We also found that the proportion of children who had good consumption of foods rich in vitamin A was discriminately concentrated among children born to mothers; with higher educational attainment, who had ANC visits during pregnancy, and who delivered at a health

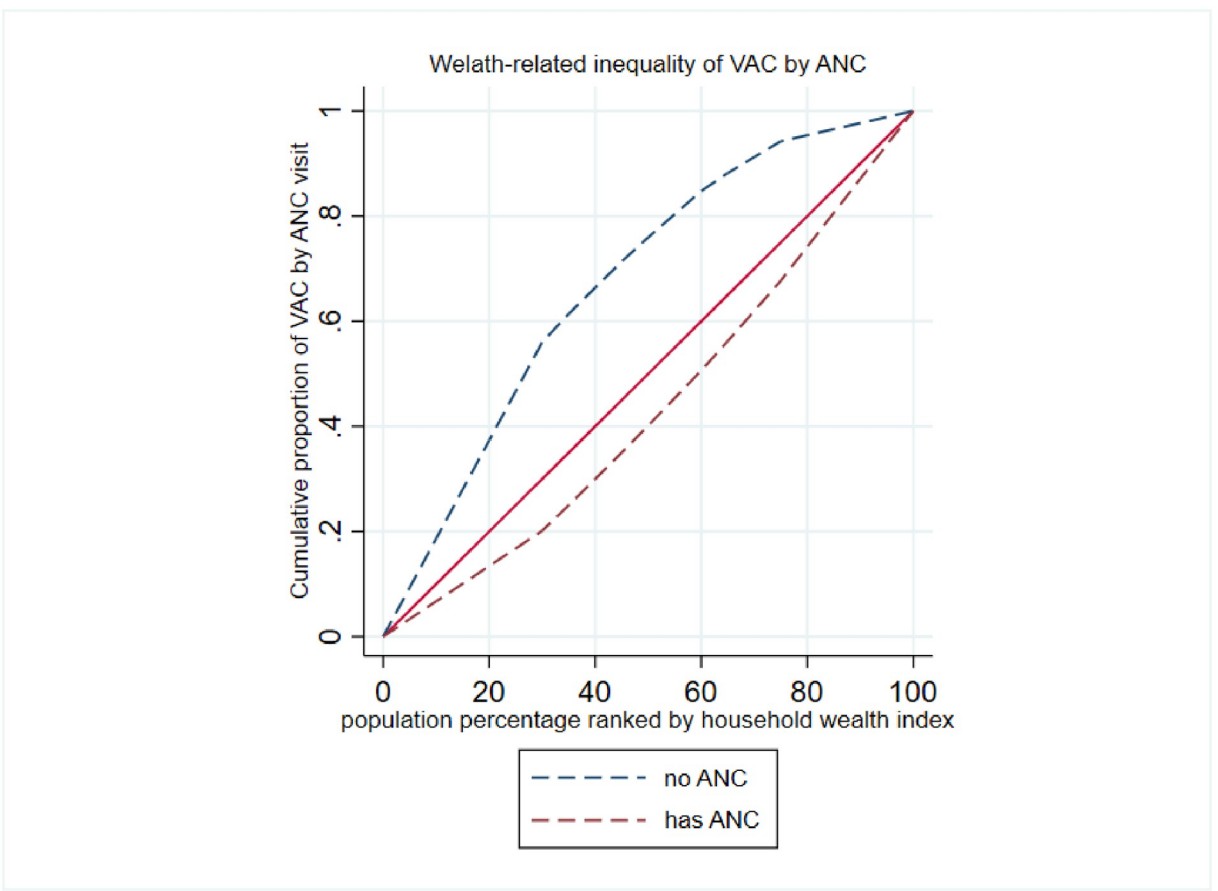

**Fig 4. Wealth related inequality to VAC among children whose mothers had ANC follow-up during pregnancy.**

institution. The geographic region and child's age were also contributed to the pro-wealth related inequalities in VAC.

As such, maternal education was one of the key contributors to the wealth related inequality in VAC where primary education contributed about 7.2% and secondary and above 17.8% to the overall concentration index. Although there has been no study assessing the contribution of maternal education to the wealth inequalities in VAC, several studies have highlighted the contribution of this factor in explaining the disparities in vitamin A supplementation, minimum dietary diversity and micro-nutrient intake among children [17, 18, 21]. The link between women education with good vitamin A consumption could be due to higher dietary knowledge [29, 44], better health literacy, dietary information-seeking behaviour, understanding, and critical thinking skills related to nutritional information of the women [46]. The other possible justification could be related to the strong link among education-occupation/income-nutrition [47–50] in which women with higher educational attainment would have better income opportunity and thus better secure the nutritional demand of their children.

In this study, we found that maternal health care utilization including the ANC visit and delivery at health facilities had made significant positive contribution to the wealth related inequality in the consumption of foods rich in vitamin A. This finding was supported with the previous studies conducted in Indonesia [21] and Ghana [42, 51].

This could be explained in that the counselling and nutrition information that the women received from the health professionals during the ANC visits would increase their health literacy and thereby help to feed their children with the recommended dietary mix including the micro-nutrients [52].

In this study, higher age category (18–23 months) of the child had positively contributed to the pro-rich inequality in VAC. This finding was in agreement with the few literatures elsewhere [17, 53]. This correlation could be explained by the fact that there has been an increase in the practice of weaning at later age of the child in most women [54, 55].

Finally, our finding noted that living in the metropolis and central region of the country has positively contributed to the overall inequality to VAC and of course in accordance with previous studies [18, 41]. The possible explanation is that children in such locations (mostly urban) belongs to better socio-economic status and thus had a higher likelihood for access to foods rich in VA compared to those children living in the periphery [18, 56].

### Strength and limitation of the study

We performed Wagstaff decomposition analysis using the most recent national survey data that enables enhanced understanding on the various factors contributing to the wealth-related inequality in consumption of foods rich in VAC among children of age 6–23 months. However, VAC is constructed based on the single 24-hour food recall during the survey, thus may not reflect the actual feeding patterns. Besides, some variables such as frequency of listening radio and watching television were not recorded in the mini EDHS 2019 which are important contributors of wealth related inequalities in VAC.

### Conclusion

The present study revealed a considerable wealth-related inequality in the VAC proportion among children of age 6–23 months in Ethiopia. Additionally, it was found that maternal education, ANC visit, place of delivery, and region contributed most towards explaining wealth-related inequality to VAC.

The finding calls for the need to multi-sectorial approaches to address the underlying contributing factors to the inequality in VAC identified in this study. Intervention strategies should prioritise addressing the significant modifiable factors identified in this study. For instance, efforts to enhance access to ANC and health institutions delivery services is highly recommended for delivering health-facility-based nutrition education and interventions in the study setting.

### Supporting information

**S1 Checklist. STROBE checklist of items for observational study used to assess the quality of our study reports.**
(DOCX)

### Acknowledgments

The authors would like to thank measure DHS for their permission to access the datasets.

## Author Contributions

**Conceptualization:** Mehari Woldemariam Merid, Fantu Mamo Aragaw, Tilahun Nega Godana, Anteneh Ayelign Kibret, Adugnaw Zeleke Alem, Melaku Hunie Asratie, Dagmawi Chilot, Daniel Gashaneh Belay.

**Data curation:** Mehari Woldemariam Merid, Fantu Mamo Aragaw, Adugnaw Zeleke Alem, Dagmawi Chilot, Daniel Gashaneh Belay.

**Formal analysis:** Mehari Woldemariam Merid, Fantu Mamo Aragaw, Dagmawi Chilot, Daniel Gashaneh Belay.

**Methodology:** Mehari Woldemariam Merid, Anteneh Ayelign Kibret, Adugnaw Zeleke Alem, Daniel Gashaneh Belay.

**Software:** Mehari Woldemariam Merid.

**Validation:** Mehari Woldemariam Merid, Fantu Mamo Aragaw, Anteneh Ayelign Kibret, Adugnaw Zeleke Alem, Melaku Hunie Asratie, Dagmawi Chilot, Daniel Gashaneh Belay.

**Visualization:** Tilahun Nega Godana, Anteneh Ayelign Kibret, Melaku Hunie Asratie, Dagmawi Chilot, Daniel Gashaneh Belay.

**Writing – original draft:** Mehari Woldemariam Merid, Tilahun Nega Godana.

**Writing – review & editing:** Mehari Woldemariam Merid, Fantu Mamo Aragaw, Tilahun Nega Godana, Anteneh Ayelign Kibret, Adugnaw Zeleke Alem, Melaku Hunie Asratie, Dagmawi Chilot, Daniel Gashaneh Belay.

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
