## [Decision Letter · Decision Letter 0]

24 Oct 2023

PONE-D-23-16225Wealth-related inequality in vitamin A rich food consumption among children of age 6-23 months in Ethiopia; Wagstaff decomposition of the 2019 mini-DHS dataPLOS ONE

Dear Dr. Mehari Woldemariam Merid,

Thank you for submitting your manuscript to PLOS ONE. After careful consideration, we feel that it has merit but does not fully meet PLOS ONE’s publication criteria as it currently stands. Therefore, we invite you to submit a revised version of the manuscript that addresses the points raised during the review process.

We look forward to receiving your revised manuscript.

Kind regards,

Jayanta Kumar Bora,PhD

Academic Editor

PLOS ONE

Journal Requirements:

- https://doi.org/10.1371/journal.pone.0281681

- https://doi.org/10.1186/s40795-022-00521-y

- https://doi.org/10.1080/16549716.2022.2040152

In your revision ensure you cite all your sources (including your own works), and quote or rephrase any duplicated text outside the methods section. Further consideration is dependent on these concerns being addressed.

Reviewers' comments:

Reviewer's Responses to Questions

**Comments to the Author**

1. Is the manuscript technically sound, and do the data support the conclusions?

Reviewer #1: No

Reviewer #2: Partly

2. Has the statistical analysis been performed appropriately and rigorously? 

Reviewer #1: Yes

Reviewer #2: Yes

3. Have the authors made all data underlying the findings in their manuscript fully available?

Reviewer #1: Yes

Reviewer #2: Yes

4. Is the manuscript presented in an intelligible fashion and written in standard English?

Reviewer #1: No

Reviewer #2: Yes

5. Review Comments to the Author

Reviewer #1: Abstract:

1. The first sentence in the introduction of your abstract is not clear. Vitamin A is the only source for what? The second sentence is not consistence with the first one. Make it clear.

2. In the introduction section of your abstract, avoid abbreviation (VAC), or use the bracket in the objective of your abstract to make it clear.

3. What is the correct term to write Wagstaff, or wag staff. Be consistent in writing.

4. In the result section of the abstract, I did not get the importance of VAC in line 38. Better if you replace it with A.

5. The recommendations in the abstract section are not informative, for instance how stakeholders improve pre-natal and institutional delivery. Specifically, what mechanisms have you recommended to improve prenatal care and institutional delivery?

Introduction:

1. I am not clear when you wrote “Inadequate micro nutrient feeding practice …..are the key determinant….. optimal growth, and development…” line 54. It needs revision

2. The authors lack to show the gap of the problem from global to local in the background section of the manuscript. So, please articulate the gap more

3. Rationally of the study is not well stated.

Method:

1. The outcome variable is not clearly written. Is “infant and young children aged……. who have taked ……..” a variable? I expect the outcome variable as “wealth-related inequality in vitamin A consumption”. Doesn’t it? If not, please write the outcome variable shortly, and then write how you measure it shortly and clearly. Plus, “have taked” is not grammatically correct.

2. From the outcome variable section, Infants and young children aged 6-23 months by themselves could not able to respond whether they have taken vitamin A-rich food or not. So, how do you know whether they take it or not? Make the readers clear about the respondents.

3. Operationalize what occupation, region, use of prenatal, use of antenatal care, current breast-feeding status, birth order, and number of under-five children in the family, family size, and household wealth index mean in your study?

4. Author missed important attributes such as media spots such as television and radio

Result:

1. Item used in figure one is not representative, including other vitamin A-rich food items

Discussion:

Strength and limitation:

1. For me, the topic which is being studied for the first time in some areas may not manifest its strength. Be specific, and write the exact strength of your paper. Avoid citation in this section.

Conclusion and recommendation:

1. The recommendation in this section is copied from the recommendation written in the abstract section. Better if you avoid duplication.

Reviewer #2: Topic was very fantastic and Everything that was stated in the manuscript was also very interesting. How ever, there is some topographical error and some issue not clear for now since your data was derived from secondary data thus why there is minor similarities with studies done before from the same data with different topic. In addition, please follow PLOS ONE Guidelines.

6. PLOS authors have the option to publish the peer review history of their article (what does this mean?). If published, this will include your full peer review and any attached files.

Reviewer #1: No

Reviewer #2: No

---

## [Author Response · Author response to Decision Letter 0]

5 Nov 2023

Date: 05 November, 2023

Rebuttal letter

Ref: PONE-D-23-16225

Title: Wealth-related inequality in vitamin A rich food consumption among children of age 6-23 months in Ethiopia; Wagstaff decomposition of the 2019 mini-DHS data

To: PLOS ONE 

Dear all,

We the authors of this manuscript are pleased to thank the journal editors and the reviewers for revising the manuscript and giving your valuable and constructive comments and suggestions that help to improve the manuscript. We have made a rigorous revision of the manuscript as per your questions and comments. We have included the point by point response in the table below framed as reviewers’ comment/question and authors’ response. The detailed revision and changes we made in the main document are prepared with track changes in the document attached separately. We expect that the revision we made will enable the manuscript to fit the journal. We are happy to receive additional revision if any that would have merit in improving the manuscript.

Editor and Reviewer comments Authors Response 

Editor comments 

Journal Requirements:

Thank you very much dear reviewer for your comments

Based on your suggestions, we re-visited the PLOS ONE submission guidelines and made amends such as section rearrangement (e.g moving the ethics statement to the methods section), renaming some subhedings (previously background now replaced by Introduction), and others in the recvised manuscript accordingly. 

- https://doi.org/10.1371/journal.pone.0281681

- https://doi.org/10.1186/s40795-022-00521-y

- https://doi.org/10.1080/16549716.2022.2040152

In your revision ensure you cite all your sources (including your own works), and quote or rephrase any duplicated text outside the methods section. Further consideration is dependent on these concerns being addressed. Thank you again dear editor 

Yes, we have noticed some overlapping texts especially in the background section and now we made intense revision and parapharased the each sections of the revised manuscript. Pages 3-5 of the introduction section and other pages of the method section.

3. Your ethics statement should only appear in the Methods section of your manuscript. If your ethics statement is written in any section besides the Methods, please move it to the Methods section and delete it from any other section. Please ensure that your ethics statement is included in your manuscript, as the ethics statement entered into the online submission form will not be published alongside your manuscript. Thank you again dear editor for your comment

As per your comment and the PLOS ONE guidline, we have moved the ethics statenment to the method section in the revised manuscript. Lines 206-212

Thank you again

Yes, we have included the captions for all the supporting documents (figures) in the revised manuscript

Reviewer 1 

Abstract:

1. The first sentence in the introduction of your abstract is not clear. Vitamin A is the only source for what? The second sentence is not consistence with the first one. Make it clear. Thank you very much dear reviwer for your comment

Yes, we made the required revision.

Line xxx

2. In the introduction section of your abstract, avoid abbreviation (VAC), or use the bracket in the objective of your abstract to make it clear. Thank you again

Sure, we made the requested change. Lines 27-29

3. What is the correct term to write Wagstaff, or wag staff. Be consistent in writing.. Thank you again

The correct one is Wagstaff and we use this consistently in the revised manuscript. 

4. In the result section of the abstract, I did not get the importance of VAC in line 38. Better if you replace it with A Thank you dear reviewer

Sure, that was typo error. We corrected it in the revised manuscript.

5. The recommendations in the abstract section are not informative, for instance how stakeholders improve pre-natal and institutional delivery. Specifically, what mechanisms have you recommended to improve prenatal care and institutional delivery? Thank you dear reviewer

In the revised manuscript, we paraphrased the recommendation as follows;

Nutritional related interventions should prioritise children of poorer households and less educated mothers. Moreover, enhancing access to ANC and health facilities delivery services through education, advocacy, and campaign programs is highly recommended in the study setting. Lines 50-53

Introduction:

1. I am not clear when you wrote “Inadequate micro nutrient feeding practice …..are the key determinant….. optimal growth, and developent…” line 54. It needs revision Thank you dear reviewer for commenting this

We re-write the statemenmt as follows;

Inadequate micro-nutrients feeding practices to infants and young children, particularly in the first 2 years of life, are the key risk factors for under-nutrition and delayed cognitive development and growth mainly in developing countries [1]. Good VA consumption refers to the intake of foods rich in VA at least one food item among the main type of food items [2]. Lines 56-59

2. The authors lack to show the gap of the problem from global to local in the background section of the manuscript. So, please articulate the gap more Thank you again

We have made a severe revision in the background section to highligjht the gap from global to local cointext as per your comment. Page 3-4 starting line 62

3. Rationally of the study is not well stated. Thank you again

In the revised manuscript, we tried to show the rationale of the study clearly.

Lines 91-100

Method:

1. The outcome variable is not clearly written. Is “infant and young children aged……. who have taked ……..” a variable? I expect the outcome variable as “wealth-related inequality in vitamin A consumption”. Doesn’t it? If not, please write the outcome variable shortly, and then write how you measure it shortly and clearly. Plus, “have taked” is not grammatically correct. Thank you dear reviewer for commenting this

Sure, we corrected the definition of the outcome variable and the grammatical error as follows;

In this study, the outcome varaiable was wealth-related inequality in consumption of foods rich in vitamin A among infants and young children aged 6–23 months living with their mother who have taken at least one food item among the seven food items……. Lines 121-131

2. From the outcome variable section, Infants and young children aged 6-23 months by themselves could not able to respond whether they have taken vitamin A-rich food or not. So, how do you know whether they take it or not? Make the readers clear about the respondents.

 Thank you dear reviewer 

Ofcurse, the questions/items themselves tells that the mothers/caregivers were asked to assess weather they fed their inants/children foods rich in VA or not. However, we added the following statements in the revised manuscript for more clarification based on your comment.

….These questions were asked to the mothers’ or the caregivers’ of the infants any time in the last 24 hours preceding the survey. Lines xxx

3. Operationalize what occupation, region, use of prenatal, use of antenatal care, current breast-feeding status, birth order, and number of under-five children in the family, family size, and household wealth index mean in your study? Thank you againg dear reviewer

We included the operational definations of the variables used in the analysis in the revised manuscript. Lines 138-158

4. Author missed important attributes such as media spots such as television and radio Thank you again for asking this.

Sure, media exposure is really important determinant factor for VAC. However, the frequency of listening to radio, watching television, and reading newsletter were not recoreded/available in the mini EDHS 2019 in which media exposure has to be generated.

Result:

1. Item used in figure one is not representative, including other vitamin A-rich food items Thank you again

Ofcaurse, there could be other sources of VA. But the DHS guidline/reports and other previous researches consistently considered the seven (egg, meat, carrot, green leafy, mango, organs, fish ) major food items as the main sources of VA.

Discussion:

Strength and limitation:

1. For me, the topic which is being studied for the first time in some areas may not manifest its strength. Be specific, and write the exact strength of your paper. Avoid citation in this section. Many thanks dear reveiewer for your comment

Baased on your comment, we have amended the strength and limitation statement in the revised manuscript. Lines 315-319

Conclusion and recommendation:

1. The recommendation in this section is copied from the recommendation written in the abstract section. Better if you avoid duplication. Thank again

We have made the required revision as per your comment. Lines 325-330

Reviewer #2: Topic was very fantastic and Everything that was stated in the manuscript was also very interesting. How ever, there is some topographical error and some issue not clear for now since your data was derived from secondary data thus why there is minor similarities with studies done before from the same data with different topic. In addition, please follow PLOS ONE Guidelines. Many thanks dear reviewer for your comments

Sure, we have went through line by line and made correction to errors and amends to as required of the revised manuscript. 

 Finaly, we the authors are very grateful to the editors and reviewers for taking your precious time and contributed significantly to the improvement of the manuscript.

 Kindest regards!

Mehari Woldemariam Merid, corresponding author

---

## [Decision Letter · Decision Letter 1]

24 Nov 2023

PONE-D-23-16225R1Wealth-related inequality in vitamin A rich food consumption among children of age 6-23 months in Ethiopia; Wagstaff decomposition of the 2019 mini-DHS dataPLOS ONE

Dear Dr. Mehari Woldemariam Merid,

Thank you for resubmitting your manuscript to PLOS ONE. After careful consideration, we feel that it has merit but does not fully meet PLOS ONE’s publication criteria as it currently stands. Therefore, we invite you to submit a revised version of the manuscript that addresses the points raised during the review process.

We look forward to receiving your revised manuscript.

Kind regards,

Jayanta Kumar Bora,PhD

Academic Editor

PLOS ONE

Reviewers' comments:

Reviewer's Responses to Questions

**Comments to the Author**

1. If the authors have adequately addressed your comments raised in a previous round of review and you feel that this manuscript is now acceptable for publication, you may indicate that here to bypass the “Comments to the Author” section, enter your conflict of interest statement in the “Confidential to Editor” section, and submit your "Accept" recommendation.

Reviewer #1: (No Response)

Reviewer #2: (No Response)

2. Is the manuscript technically sound, and do the data support the conclusions?

Reviewer #1: Yes

Reviewer #2: Yes

3. Has the statistical analysis been performed appropriately and rigorously? 

Reviewer #1: Yes

Reviewer #2: I Don't Know

4. Have the authors made all data underlying the findings in their manuscript fully available?

Reviewer #1: Yes

Reviewer #2: Yes

5. Is the manuscript presented in an intelligible fashion and written in standard English?

Reviewer #1: Yes

Reviewer #2: Yes

6. Review Comments to the Author

Reviewer #1: 2nd revision

The authors made revisions, but some points/ revisions are not satisfactory. So, I suggest the authors to look at the following points before the publication of the manuscript

1. The gap in the literature on the topic could not be the only reason to justify the importance of the research. Therefore, the justification is still not satisfactory.

2. Under the operationalization section, citation is required, for the variables, at the end of the sentence.

3. Authors still do not understand the difference between media exposure, and frequency of listening, watching, and reading of Radio, TV, and newspaper respectively. Many researchers have used the radio and TV to measure media exposure and literature, published based on 2019 mini demographic and health survey, shows that media exposure has records in 2019 mini demographic health survey.

Reviewer #2: My Comments and concerns on Manuscript

Entitled as ‘’Wealth-related inequality in vitamin A rich food consumption among children of age 6- 23 months in Ethiopia; Wagstaff decomposition of the 2019 mini-DHS data’’.

Short title: - written as “Inequalities in iron rich food consumption” which is not related with your main topic. Omit Iron from here.

Abstract:-

‘’Foods rich in vitamin A (VA) is the only source since it cannot be made in the human body’’. The only source of what? This statement does not give any sense and unclear words. Make it clear and specific.

Introduction:

Lack of flow of idea and redundant ones…..Check Paragraph 3, line 7 up to 11.

Methods and Materials

Data sources and populations

Even though the topic is quite different from previous study done in Ethiopia, ‘’spatial distribution of vitamin A rich foods intake and associated factors among children aged 6–23 months in Ethiopia: spatial and multilevel analysis of 2019 Ethiopian mini demographic and health survey’’, However Data sources and populations some similarities. Make it different from previous studies that were published elsewhere.

Variables of the study:

You interviewed the child taken fish any time in the last 24 hours preceding the interview and considered as good consumption of foods rich in vitamin A coded as “1”, otherwise, no consumption coded as “0”. So, could you believe that such like judgment is scientifically sound? Since your data classify based on 24 hours consumption only.

Let’s raise some concerns.

1. How do you know that this child was taken fish/VAC always or not?

2. How do you know if this child was only taken fish for the first and last at those times?

3. Assume that this child was consumed VAC/fish always, but probably did not taken any fish/VAC within 24 hours, what was your judgment?

4. What about the other outcome variables listed 1-6?

Add operational definitions

Data management and analysis

‘’Its value ranges from − 1 and + 1, where C = 0 shows perfect equality while C < 0 indicates VAC is disproportionately concentrated among the poor (pro-poor inequality, and C>0 means VAC is disproportionately concentrated among the poor (pro-rich inequality)’’.

From this interpretation both C<0 & C>0 indicates it was almost similar i.e. ‘’VAC is disproportionately concentrated among the poor (pro-poor inequality’’. Why you need to state as different interpretation? Again how it was similar? Check it again and make correction.

Results

Description of Vitamin A consumption by the maternal and child characteristics:-You put table 1 caption but no table is found at the place. It’s better if you put the description and its table together

Decomposition of wealth-related inequality in VAC:- You put table 2 caption but no table is found at the place. It’s better if you put the description and its table together.

Discussion

Some of variables were not discussed well and some doesn’t have adequate justification.

Again at the end your discussion you put single line i.e. ’’In this study, only 39%’’ statements which is not clear and not discussed. Make it clear.

Conclusion

Conclusion at end was not related with the conclusion you stated in the Abstract section (Region is contributing factor for VAC)….check it again.

Where is your recommendation?

Ethical consideration

Write the Ethical consideration at the beginning of Methods and materials.

List of Tables

Table 1 has some discordant i.e. Women’s age, Marital status, Household wealth index, ANC Visit and Children Age in month was 1498 but your total study participants was 1497. How it could happen? Please, check it attentively.

Again on your Table 1:-You state one variable as “Highest education”. What it mean? Please, change into Mother educational status. Again don’t use any abbreviation like

U-5, ANC, PNC and others.

Table 2: The topic was about VAC among children aged 6–23 months but it was mixed with mother characteristics like Age, Marital status, Education, Residence, etc. So update this table description.

List of Figures

No need to write as list of figures at the end however, it’s better if you only put figures with caption.

Labeling and description of Figure 1 was not written well.

7. PLOS authors have the option to publish the peer review history of their article (what does this mean?). If published, this will include your full peer review and any attached files.

Reviewer #1: No

Reviewer #2: **Yes: **Fedasan Alemu Abdi

---

## [Author Response · Author response to Decision Letter 1]

2 Jan 2024

Date: 2nd of Jan 2024

Rebuttal letter

Ref: PONE-D-23-16225R1

Title: Wealth-related inequality in vitamin A rich food consumption among children of age 6-23 months in Ethiopia; Wagstaff decomposition of the 2019 mini-DHS data

To: PLOS ONE 

Dear all,

We the authors of this manuscript are pleased to thank the journal editors and the reviewers for revising the manuscript and giving your valuable and constructive comments and suggestions that help to improve the manuscript. We have made a rigorous revision of the manuscript as per your questions and comments. We have included the point by point response in the table below framed as reviewers’ comment/question and authors’ response. The detailed revision and changes we made in the main document are prepared with track changes in the document attached separately. We expect that the revision we made will enable the manuscript to fit the journal. We are happy to receive additional revision if any that would have merit in improving the manuscript.

Editor and Reviewer comments Authors Response 

Reviewer #1 comments 

2nd revision

The authors made revisions, but some points/ revisions are not satisfactory. So, I suggest the authors to look at the following points before the publication of the manuscript

1. The gap in the literature on the topic could not be the only reason to justify the importance of the research. Therefore, the justification is still not satisfactory. Thank you very much dear reviewer for your comment

We included the justification/rationle for counducting the current study throughout writing the introduction e.g. lines 90-96

Also, we added a justification statement in the revised manuscript. Lines 100-102

2. Under the operationalization section, citation is required, for the variables, at the end of the sentence. Thank you again

We have used common citation, particularly the DHS guide, for defining the variables operationally. This has already been cited at the beginning and we opt to keep this for the sake of not making redendecy by citing at the end of each staements. However, we cited a specific literatures to some variables as required. Page 7

3. Authors still do not understand the difference between media exposure, and frequency of listening, watching, and reading of Radio, TV, and newspaper respectively. Many researchers have used the radio and TV to measure media exposure and literature, published based on 2019 mini demographic and health survey, shows that media exposure has records in 2019 mini demographic health survey. Thank againg for raising this issue.

Ofcourse, some researchers may consider media exposure solely by aggregating the presence of television/radio in the household. However, the mere presence of these does not actually measure the media exposure/use. The correct approach to measure the media exposure is by aggregating the frequency of watching television, listening radio, reading newspapers. These data are however not collected in the 2019 mini EDHS. Theferore, we prefer to keep our analysis as such and acknoweledege as limitation about this. Lines 320--323

Reviewer #2 commnets Author Response 

Short title: - written as “Inequalities in iron rich food consumption” which is not related with your main topic. Omit Iron from here. Thank you again dear reviewer

Sure, we corrected it and replaced it by vitamin A.

Abstract:-

1. ‘’Foods rich in vitamin A (VA) is the only source since it cannot be made in the human body’’. The only source of what? This statement does not give any sense and unclear words. Make it clear and specific.

Introduction: Thank you again dear reviewer for your comment

We have made clarification to the statement in the revised manuscript as follows; 

Vitamin A (VA) cannot be made in the human body and thus foods rich in VA are the only sources of vitamin A for the body.

2. Lack of flow of idea and redundant ones…..Check Paragraph 3, line 7 up to 11. Thank you again

Based on your comment we have paraphrased the paragraph in the revised manuscript. Lines 70-75

Methods and Materials

Data sources and populations

1. Even though the topic is quite different from previous study done in Ethiopia, ‘’spatial distribution of vitamin A rich foods intake and associated factors among children aged 6–23 months in Ethiopia: spatial and multilevel analysis of 2019 Ethiopian mini demographic and health survey’’, However Data sources and populations some similarities. Make it different from previous studies that were published elsewhere. Thank you again

The data source and the population used for the current study was from the mini EDHS 2019 data which is the publicly available at (https://www.dhsprogram.com. Due to the publicly accessibility nature of the data and the same methodology and population discreption guiline outlined in the DHS, there could be some similarities across different articles done using this data. However, we have tried to paraphrase and make different to the previous studies. 

Variables of the study:

2. You interviewed the child taken fish any time in the last 24 hours preceding the interview and considered as good consumption of foods rich in vitamin A coded as “1”, otherwise, no consumption coded as “0”. So, could you believe that such like judgment is scientifically sound? Since your data classify based on 24 hours consumption only. Thank you very much dear reviwer for your comment

Indeed, the gold standard method to deal with the level of vitamins and minerals in the body is by measuruing the blood level of the nutrients or by using some dietetric assessment approaches. However, the present study is based on the secondary survey data on large population where the other dietary assessment methods would not be feasible. For this reason, the data collection method was based on the 24-hour food recall questionnaire, the commonly used method in national dietary survey studies (Caswell et al., 2015, Salvador Castell et al., 2015).

We actually, acknowledge for the presence of the possible recall bias in the limitation section of the manuscript. Lines 320-323

3.1. How do you know that this child was taken fish/VAC always or not? Thank you again

Though it is as stated in the previous response, it is assumed that the 24-hour food recall assessment could reflect the usual household feeding practice.

3.2. How do you know if this child was only taken fish for the first and last at those times? Thank you again

Same to the above

3.3. Assume that this child was consumed VAC/fish always, but probably did not take any fish/VAC within 24 hours, what was your judgment? Thank you dear reviewer

Similarly, this is actually a survey data that countries worldwide used to assess health staus, health indicators, and dietary patterns at population level using the most feasible approach every five years. The assessment is thought to represent the average usual nutritional habit of the household. And we the researchers have no control over the data collection method in such reglar national surveys. We however produce evidences based on the available data by interpreting the findings cautiously and by taking in account all the possible limitations.

3.4. What about the other outcome variables listed 1-6? Thank you dear reviewer

The outcome varaiable, consumption foods rich in vitamin A, was assessed through the 24-hour food recall assessment questionnaire as described in the previous reseponses. 

4. Add operational definitions Thank you again

We have included the operational definations. Page 7

Data management and analysis

1. ‘’Its value ranges from − 1 and + 1, where C = 0 shows perfect equality while C < 0 indicates VAC is disproportionately concentrated among the poor (pro-poor inequality, and C>0 means VAC is disproportionately concentrated among the poor (pro-rich inequality)’’.

From this interpretation both C<0 & C>0 indicates it was almost similar i.e. ‘’VAC is disproportionately concentrated among the poor (pro-poor inequality’’. Why you need to state as different interpretation? Again how it was similar? Check it again and make correction. Thank you dear reviewer for asking this

That was a typo-error in which the word “poor” would not have been used when C was greater than 0. We made correction in the revised manuscript where the word poor was replaced by rich and interpreted as follows;

Its value ranges from − 1 and + 1, where C = 0 shows perfect equality while C < 0 indicates VAC is disproportionately concentrated among the poor (pro-poor inequality), and C>0 means VAC is disproportionately concentrated among the rich (pro-rich inequality) (Wagstaff et al., 2007).

Results 

1. Description of Vitamin A consumption by the maternal and child characteristics:-You put table 1 caption but no table is found at the place. It’s better if you put the description and its table together Thank you dear reviewer

There is table 1 and its located at the end of the manuscript (page 20) as per the format of the Plos One manuscript writng guideline.

2. Decomposition of wealth-related inequality in VAC:- You put table 2 caption but no table is found at the place. It’s better if you put the description and its table together. Thank you again dear reviewer

Same to the above, table 2 its located at the end of the manuscript (page 21) as per the format of the Plos One manuscript writng guideline.

Discussion

1. Some of variables were not discussed well and some doesn’t have adequate justification. Thank you dear reviewer

We made a revision to the variables and made some more ellborated discussion points to the variables and added possible and plausible justification points as well. Page 13. We welcome further comments specific to certain variables, if any so that we can do more.

2. Again at the end your discussion you put single line i.e. ’’In this study, only 39%’’ statements which is not clear and not discussed. Make it clear. Thank you dear reviewer

We were unable to find a text ’’In this study, only 39%’’ in the discussion. We might have corrected it previously. We are still happy to receive if any.

 Conclusion 

1. Conclusion at end was not related with the conclusion you stated in the Abstract section (Region is contributing factor for VAC)….check it again. Thank you dear reviewer

We have revised for the consistency of the conclusion statements in the abstract section and at the end of the manuscript and included the variable region.

2. Where is your recommendation? Thank you dear reviewer

We have already included the recommendation as follows; 

The finding calls for the need to multi-sectorial approaches to address the underlying contributing factors to the inequality in VAC identified in this study. Intervention strategies should prioritise addressing the significant modifiable factors identified in this study. For instance, efforts to enhance access to ANC and health institutions delivery services is highly recommended for delivering health-facility-based nutrition education and interventions in the study setting. Lines 329-334

Ethical consideration

1. Write the Ethical consideration at the beginning of Methods and materials. Thank you dear reviewer

As per the Plos One journal guideline, the Ethical consideration should be written at the end of the methods section or immediately before the result section. https://journals.plos.org/plosone/s/submission-guidelines

List of Tables

1. Table 1 has some discordant i.e. Women’s age, Marital status, Household wealth index, ANC Visit and Children Age in month was 1498 but your total study participants was 1497. How it could happen? Please, check it attentively. Thank you dear reviewer

Such discordants in the number of observations in each categories of the variables had happened when we made an approximation to the nearest digit. Based on your review, we have re-analysed the tabulation and made correction in the updated version of the manuscript. Table 1 pages 19-21

2. Again on your Table 1:-You state one variable as “Highest education”. What it mean? Please, change into Mother educational status. Again don’t use any abbreviation like

U-5, ANC, PNC and others. Thank you dear reviewer

Based on your recommentdation, we have replaced the variable “Highest education” by mother educational status in the table and other sections of the manuscript. Regarding the abbreviation like

U-5, ANC, PNC, we have indicated a foot note under that table to reduce the inside table bulkiness and to ease understanding as;

ANC-antenatal care, BF-breast feeding, C –concentration index, PNC-postnatal care.

3. Table 2: The topic was about VAC among children aged 6–23 months but it was mixed with mother characteristics like Age, Marital status, Education, Residence, etc. So update this table description.

 Thank you dear reviewer

Sure, we have update the caption/table discreption as follows in the revised manuscript.

Table 2: Decomposition of the concentration index of wealth-related inequalities to VAC attributable to maternal and child factors among children aged 6–23 months in Ethiopia; mini EDHS 2019.

List of Figures

4. No need to write as list of figures at the end however, it’s better if you only put figures with caption. Thank you again 

We made this to meet the Plos One journal manuscript writing guideline. 

5. Labeling and description of Figure 1 was not written well. Thank you dear reviewer for commenting this

For better understanding, we have clarified the caption of figure 1 as follows;

Figure 1: Proportions of the consumption of food items rich in VA by children of age 6-23 months in Ethiopia

 Finaly, we the authors are very grateful to the reviewers for taking your precious time and contributed significantly to the improvement of the manuscript.

 Kindest regards!

Mehari Woldemariam Merid, corresponding author 

CASWELL, B. L., TALEGAWKAR, S. A., DYER, B., SIAMUSANTU, W., KLEMM, R. D. & PALMER, A. C. 2015. Assessing Child Nutrient Intakes Using a Tablet-Based 24-Hour Recall Tool in Rural Zambia. Food Nutr Bull, 36, 467-80.

SALVADOR CASTELL, G., SERRA-MAJEM, L. & RIBAS-BARBA, L. 2015. What and how much do we eat? 24-hour dietary recall method. Nutr Hosp, 31 Suppl 3, 46-8.

WAGSTAFF, A., O'DONNELL, O., VAN DOORSLAER, E. & LINDELOW, M. 2007. Analyzing health equity using household survey data: a guide to techniques and their implementation, World Bank Publications.

---

## [Decision Letter · Decision Letter 2]

3 Apr 2024

Wealth-related inequality in vitamin A rich food consumption among children of age 6-23 months in Ethiopia; Wagstaff decomposition of the 2019 mini-DHS data

PONE-D-23-16225R2

Dear Dr. Mehari Woldemariam Merid,

We’re pleased to inform you that your manuscript has been judged scientifically suitable for publication and will be formally accepted for publication once it meets all outstanding technical requirements.

Kind regards,

Jayanta Kumar Bora,PhD

Academic Editor

PLOS ONE

Additional Editor Comments (optional):

Reviewers' comments:

Reviewer's Responses to Questions

**Comments to the Author**

1. If the authors have adequately addressed your comments raised in a previous round of review and you feel that this manuscript is now acceptable for publication, you may indicate that here to bypass the “Comments to the Author” section, enter your conflict of interest statement in the “Confidential to Editor” section, and submit your "Accept" recommendation.

Reviewer #2: All comments have been addressed

2. Is the manuscript technically sound, and do the data support the conclusions?

Reviewer #2: Yes

3. Has the statistical analysis been performed appropriately and rigorously? 

Reviewer #2: Yes

4. Have the authors made all data underlying the findings in their manuscript fully available?

Reviewer #2: Yes

5. Is the manuscript presented in an intelligible fashion and written in standard English?

Reviewer #2: Yes

6. Review Comments to the Author

Reviewer #2: Thanks dear Authors! Almost all my concerns was addressed. I don't have any comments. You can proceed it.

7. PLOS authors have the option to publish the peer review history of their article (what does this mean?). If published, this will include your full peer review and any attached files.

Reviewer #2: **Yes: **Fedasan Alemu Abdi

---

## [Editor Report · Acceptance letter]

27 Sep 2024

PONE-D-23-16225R2 

PLOS ONE

Dear Dr. Merid, 

I'm pleased to inform you that your manuscript has been deemed suitable for publication in PLOS ONE. Congratulations! Your manuscript is now being handed over to our production team.

Kind regards, 

on behalf of

Dr. Jayanta Kumar Bora 

Academic Editor

PLOS ONE